# A Study on the Resolution and Depth of Focus of ArF Immersion Photolithography

**DOI:** 10.3390/mi13111971

**Published:** 2022-11-14

**Authors:** Jungchul Song, Chae-Hwan Kim, Ga-Won Lee

**Affiliations:** 1Office of Nano Convergence Technology, National NanoFab Center, Daejeon 34141, Korea; 2Division of Electronics Engineering, Chungnam National University, Yusung-gu, Daejeon 34134, Korea

**Keywords:** photolithography, ArF immersion, DOF, resolution, PR thickness

## Abstract

In this study, the resolution and depth of focus (DOF) of the ArF immersion scanner are measured experimentally according to numerical aperture (*NA*). Based on the experiment, the theoretical trade-off relationship between the resolution and depth of focus can be confirmed and *k*_1_ and *k*_2_ are extracted to be about 0.288 and 0.745, respectively. Another observation for a problem in small critical dimension realization is the increase in line width roughness (LWR) according to mask open area ratio. To mitigate the trade-off problem and critical dimension variation, the photoresist thickness effect on depth of focus is analyzed. Generally, the photoresist thickness is chosen considering depth of focus, which is decided by *NA*. In practice, the depth of focus is found to be influenced by the photoresist thickness, which can be caused by the intensity change of the reflected ArF light. This means that photoresist thickness can be optimized under a fixed *NA* in ArF immersion photolithography technology according to the critical dimension and pattern density of the target layer.

## 1. Introduction

In accordance with Moore’s law, semiconductor technology nodes have developed rapidly in the direction of reduced size. This technological development is supported by the ongoing evolution of photolithography [1]. A fine patterning process is essential to produce a small-sized chip. For advanced technology, the photolithography process should have good resolution, and the wavelength of the light source should be reduced [2]. KrF 248 nm and ArF 193 nm excimer laser light sources and EUV (extreme ultraviolet) 13 nm via G-line 436 nm, i-line 365 nm, and halogen lamp light sources have been developed [3]. With the advanced light source, researchers have continued to increase the numerical aperture (*NA*) to improve the resolution in ArF light sources, focusing on enlarging the diameter of the lens. However, the focal error can increase in the lens manufacturing process, and thus the lens aberration becomes larger when the size of the lens increases [4]. To improve the *NA* without increasing the size of the lens, a method of putting water between the lens and the photoresist has been adopted called an ArF immersion system [2,5,6]. The ArF immersion system is known to be used for a resolution with a critical dimension of 38 nm line width. The ArF immersion system has a resolution of 42 nm critical dimension based on the line and space pattern, and can achieve a resolution of up to 38 nm with optical materials of diffractive optical elements (DOE) and technologies such as phase shift mask (PSM) and off-axis illumination (OAI) [7,8,9].

In this paper, the resolution and depth of focus of ArF immersion scanners are extracted based on the experiments according to *NA*. To optimize the lithography process, especially in photoresist, the results are analyzed varying the line and space pattern size. It is confirmed that the reflection of the light source has a major influence on the depth of focus, profile of the pattern, and process margin.

## 2. Materials and Methods

### 2.1. Resolution and Depth of Focus according to *NA*

The ArF immersion scanner of ASML’s XT-1900Gi model and the track of TEL’s Lithius Pro-I are used for the lithography experiment, with a p-type silicon wafer. In the ArF immersion process, two types of bottom anti-reflective coating (BARC) with different refractive indices were used on the underlayer of the photoresist. Reflective topographic substrates will cause changes in the swing ratio of the resist and generate linewidth variations, standing waves, and reflective notching in the resist profiles. The application of BARC has been found to be an effective tool to reduce or eliminate substrate reflection [10]. The photoresist is a 900 Å thick positive immersion photoresist. As the fixed conditions of the ArF immersion scanner, the diffractive optical element (DOE) is Dipole Y 35 and the sigma out–in value is 0.98−0.82. *NA* of the projection optics, defined by the following equation;
(1)NA=nsinθ,
where *n* is the refractive index of the medium of the focal region and θ is the maximum incidence angle. Because n is generally larger in a liquid than in the air, the *NA* of the ArF immersion system can be higher than that of a conventional dry system with the same incidence angle θ [11]. In this experiment, *NA* can be adjusted from 0.85 to 1.35. Masks having a line and space ratio of 1:1, 1.5:1, 2:1, and 3:1 with various CDs are used.

Theoretically, the resolution of the optical exposure system is given by the following formula:(2)$$\mathrm{Resolution}=k_{1}\frac{\lambda }{NA}=k_{1}\frac{\lambda }{n\sin\theta }\;,$$
where $k_{1}$ is a constant and λ is the wavelength of the light source. The resolution is enhanced by factor n, reducing λ to an effective wavelength of λ/n, which is the advantage of the immersion system as mentioned before. In Equation (2), *NA* must be large for fine resolution and 38 nm can be obtained at an *NA* of 1.35, which is the maximum value of the immersion scanner in this experiment. At a minimum *NA* of 0.85, the best resolution is 70 nm. Considering that the maximum *NA* that can be set in an ArF dry system is 0.85, the resolution can be improved from 70 nm to 38 nm by the immersion system.

Figure 1a is a CD-SEM of a 1:1 line and space pattern with a 38 nm critical dimension after photoresist patterning. Based on the CD SEM, the resolutions are extracted experimentally according to *NA*, as shown in Figure 1b with *k*_1_ = 0.288. The exposure energy required for photoresist patterning is also extracted at each condition, illustrating that the required amount of energy increases for the smaller pattern. This is one of the severe problems that appear when the critical dimension is much smaller than the wavelength of the light source. In this small critical dimension range, the resolution can be affected by the mask pattern density, which will be analyzed in a later section.

In case of depth of focus, it can be expressed as the following equation [12];
(3)$$\mathrm{Depth}\;\mathrm{of}\;\mathrm{focus}\;=k_{2}\frac{\lambda }{NA^{2}},$$
where $k_{2}$ is a constant. For a good resolution, *NA* must be increased and λ minimized by the aforementioned formula, Equation (2). In this case, the depth of focus is deteriorated by Equation (3). To demonstrate the relationship between resolution and depth of focus according to the *NA*, depth of focus is extracted where photoresist thickness is enlarged to 2400 Å to find out the depth of focus experimentally.

Figure 2a shows the extracted depth of focus at each *NA* for the minimum critical dimension from 38 nm to 70 nm. Figure 2b is the critical dimension FE-SEM of the pattern. As mentioned before in Equations (2) and (3), it can be seen that the depth of focus is decreased as *NA* increases, showing clearly the trade-off relationship between the resolution and depth of focus [13]. Point A with *NA* 1.35 has 38 nm and Point B with *NA* 1.2 has 44 nm top critical dimension, which is the maximum resolution at each *NA*. Based on the experiment, *k*_2_ is extracted to be 0.745.

As in Figure 2a, the depth of focus is under 50 nm for 38 nm critical dimension with *NA* of 1.35. One of the key parameters that impact the final critical dimension is the photoresist thickness. In this case, the photoresist thickness should be controlled to be thinner than the depth of focus. In the microfabrication process, however, the minimum critical dimension can be larger than 38 nm in another layer and this thin photoresist thickness can cause a large variation in the critical dimension. In practice, Figure 3a presents the experimental results of the depth of focus according to *NA* at 100 nm critical dimension with FESEM of the pattern in Figure 3b. It can be seen that photoresist thickness can be increased to about 75 nm at 1.35 *NA* in the layer which has a minimum critical dimension of 100 nm. The results show that critical dimension control can be possible with optimized photoresist thickness in each layer.

### 2.2. Resolution according to Mask Open Area Ratio

As aforesaid in Figure 1b, the required exposure energy for photoresist patterning varies depending on the critical dimension and increases in the pattern of 100 nm or less where the critical dimension is much smaller than the wavelength of the light source. In this case, the resolution can be affected by the mask pattern density. The photolithography mask is divided into a blocking (Cr) area for light incidence and an open (quartz) area through which light is transmitted. Since using a positive photoresist, the Cr area of the mask forms a pattern line, and the open area becomes a space in photoresist on the substrate. Generally, the line critical dimension of the pattern decreases as the intensity value increases [14].

In the previous experiment, the Cr and quartz area ratio of the mask (mask open area ratio) are fixed at 1:1. To analyze the effect of mask pattern density on resolution, an exposure experiment was conducted on the same positive photoresist, varying mask open area ratio with 1:1, 1.5:1, 2:1, and 3:1.

Figure 4a shows that the line width roughness (LWR) estimated by CD-SEM varies depending on the mask open area ratio, with *NA* = 1.35. With the decrease in semiconductor device dimensions, LWR becomes one of the most important sources of device variation and thus needs to be controlled below 2 nm for the future technological nodes of the semiconductor roadmap [15]. When the value of the LWR is 3.5 or more, a stable subsequent process cannot be performed. Thus, LWR should be controlled to be below the dashed line in Figure 4a. Furthermore, according to the International Technology Roadmap for Semiconductors (ITRS), LWR should not exceed 8% of the gate line width to limit the impact on device performance [15]. When the mask open area ratio is 1:1, this constraint in LWR is satisfied in all critical dimension ranges, but in other ratios, LWR is tolerable near the 100 nm critical dimension range. Figure 4b shows that as the mask open area ratio (Cr area ratio) increases, the required energy increases, which explains the increase in LWR and the limit of immersion ArF lithography.

## 3. Results and Discussion

### Photoresist Thickness and Depth of Focus

Theoretically, the depth of focus varies depending on *NA*, which has a direct effect on the thickness of the photoresist pattern [16,17]. However, the amount of reflected light source depends on the thickness of the photoresist and the refractive index of the lower film, and this can cause reinforcement and offset interference within the photoresist. Due to this effect of the reflection, the depth of focus is affected by photoresist thickness and the lower film. Reinforcement interference by reflection corresponds to the $k_{2}$ constant of Equation (3). When the photoresist thickness becomes larger than the depth of focus, the influence of the reflected light is reduced, and thus the interference is also reduced. That is, the $k_{2}$ constant decreases. Conversely, as the photoresist thickness and depth of focus get closer, it can be seen that the depth of focus increases due to the increase in $k_{2}$ constant.

Selecting the photoresist thickness optimized for the depth of focus changed according to the pattern density and critical dimension is a very important factor in photolithography. In the experiment, photoresist thickness is varied to 1100 Å, 2100 Å, 2300 Å, and 3300 Å by adjusting the spin RPM, and the depth of focus was extracted according to the photoresist thickness.

Figure 5a shows the depth of focus according to photoresist thickness at *NA* of 1.35, 1.0, and 0.85 (critical dimension 38 nm, 60 nm, 80 nm). Figure 5b shows that the entire photoresist is active beyond the threshold intensity and exposed to the lower film when the depth of focus is sufficiently larger than the photoresist thickness. However, it can be seen that as photoresist thickness increases, the depth of focus rapidly decreases after a certain thickness. This occurs between 2100 Å and 2300 Å at *NA* 1.35, and between 2300 Å and 3300 Å at *NA* 1.0. This is because when photoresist thickness becomes too thick, there is no interference by reflection and thus only depth of focus by the incident light source is left. Therefore, for a good pattern profile, photoresist thickness should be thin enough, which deepens in a small critical dimension. Generally, when determining photoresist thickness for the target critical dimension, the *NA* suitable for the resolution should be selected first. Because the depth of focus varies according to *NA*, the photoresist thickness must be selected according to the depth of focus. Based on the experiment, however, the depth of focus is shown to depend on the photoresist thickness at which the depth of focus is maximized by the reflection interference effect. This optimized thickness seems to become larger as the critical dimension increases. Therefore, even in *NA* = 1.35, the photoresist thickness has a margin to increase in the layer where the minimum critical dimension is larger, lessening the LWR problem.

## 4. Conclusions

Currently, products based on 7 nm process technology by EUV photolithography are being released in the semiconductor market. However, more than 30% of the manufacturing process still requires the ArF immersion process, and thus the ArF immersion system is still important. Through the experiments in this study, the resolution and depth of focus according to *NA* and the effects of mask open area ratio and photoresist thickness were analyzed using the ArF immersion system. The depth of focus is decreased as *NA* increases, showing clearly the trade-off relationship between the resolution and depth of focus. That is, the *NA* is chosen depending on the critical dimension of a layer in the microfabrication process, and decides the depth of focus. Another problem in small critical dimension realization is the increase in LWR, shown in resolution according to the mask open area ratio. To mitigate the trade-off problem and overcome the limit of resolution due to the mask open area ratio, the depth of focus is extracted according to photoresist thickness. Practically, the depth of focus is shown to vary according to the photoresist thickness by the interference effect. Therefore, it is important to select the appropriate photoresist thickness according to the critical dimension and pattern of the target layer, which would affect the process margin in the photolithography and the subsequent process.

## Figures and Tables

**Figure 1 micromachines-13-01971-f001:**
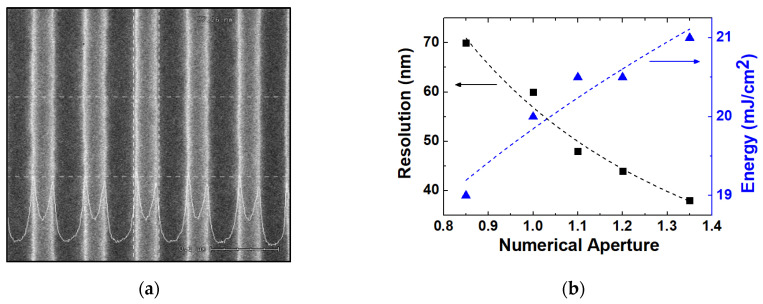
(**a**) Results of CD-SEM of 38 nm line and space pattern after photoresist development (top view) and (**b**) the extracted resolution according to *NA* where the optimized energy of light source is also illustrated. Energy density to implement the maximum resolution of the *NA* value. (Here, the 1:1 line and space pattern is used.)

**Figure 2 micromachines-13-01971-f002:**
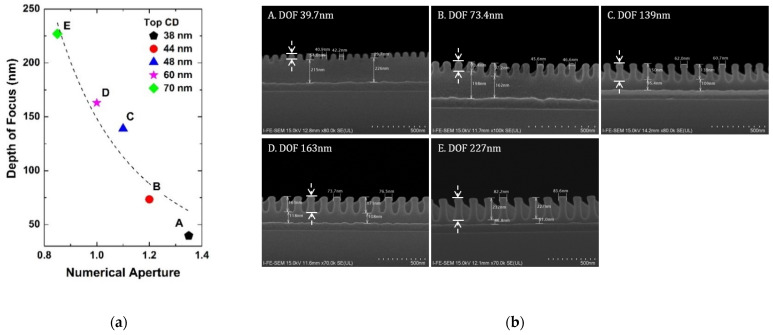
(**a**) Extracted results of the depth of focus according to *NA*. At each *NA*, the critical dimension is the best (maximum) resolution; (**b**) FE-SEM of 1:1 line and space pattern (cross view) with the best resolution at each *NA* condition. Here, the thickness of the photoresist is enlarged to 2400 Å to find out the depth of focus experimentally. The dashed arrows indicate the depth of focus.

**Figure 3 micromachines-13-01971-f003:**
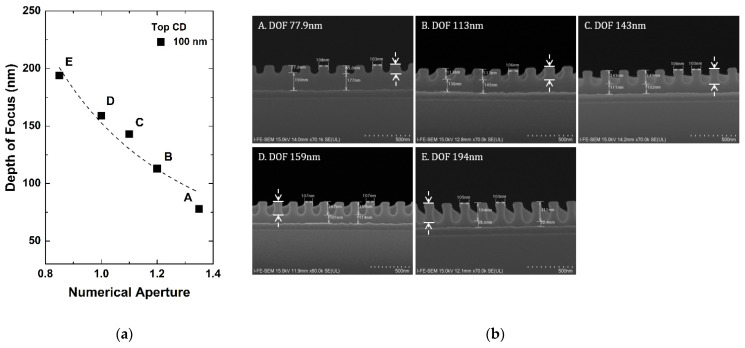
(**a**) Extracted depth of focus according to *NA* in 1:1 line and space pattern with 100 nm, and (**b**) FE SEM analysis of the pattern (cross view). Here, the photoresist thickness is enlarged to 2400 Å to find out the depth of focus experimentally. The dashed arrows indicate the depth of focus.

**Figure 4 micromachines-13-01971-f004:**
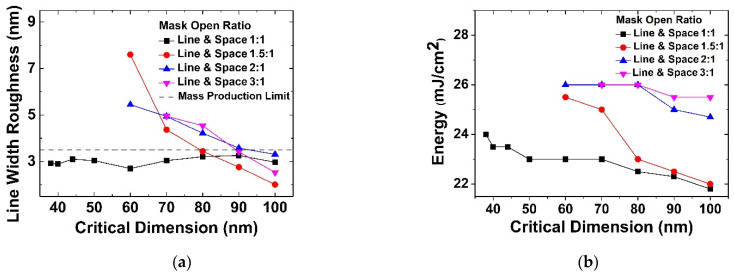
Experimental results on (**a**) LWR, and (**b**) expose energy according to mask open area ratio with fixed *NA* of 1.35.

**Figure 5 micromachines-13-01971-f005:**
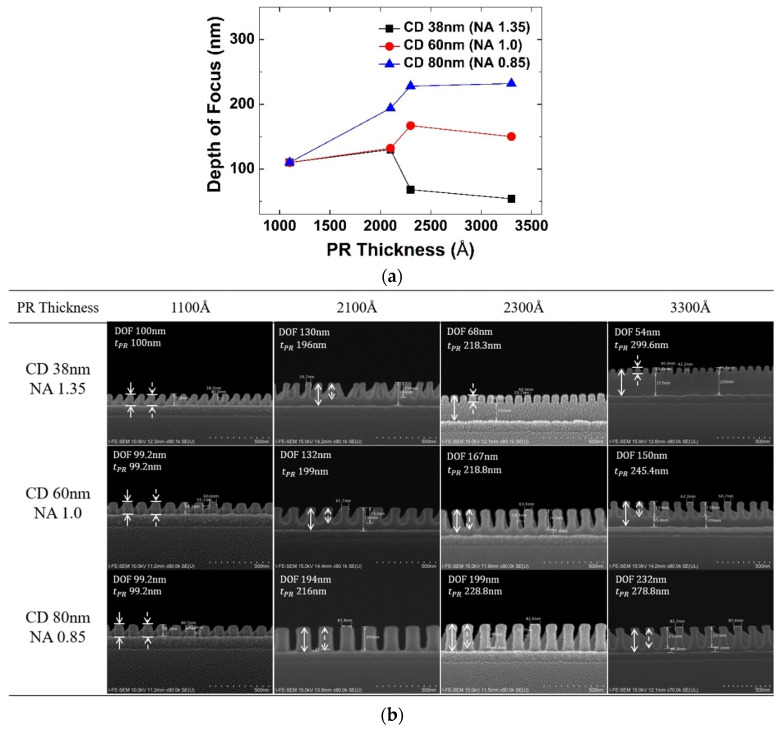
(**a**) Changes in depth of focus according to photoresist thickness, (**b**) result of FE-SEM changes in depth of focus according to photoresist thickness. The dashed arrows indicate the depth of focus and the solid arrows indicate the thickness of PR.

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
