# Peer review of "A Study on the Resolution and Depth of Focus of ArF Immersion Photolithography"

_micromachines, 2022, doi:10.3390/mi13111971_

Round 1

Reviewer 1 Report

I am not a specialist in the field of photolithography, but, if I understand correctly, this manuscript addresses the subject of achievable resolution as a function of "lens" aperture size, e.g. theta in equation 1, and the "open aperture ratio" of the optical mask being used.  The work appears to be well performed and for the fraction of readers working within this specific field, the description is probably understandable.  However, because the text abounds with abbreviations/jargon that may be standard within the authors' laboratory, perhaps even to the entire photolithography community, for the general reader the information is difficult, if not impossible, to extract without considerable additional reading of the literature.  For example, to the general reader the term "depth of focus" implies how deep within the substrate focus is achieved.  But, doing a quick literature search I find that even within the field, this term has different interpretations (see C.A. Mack,  proceedings of  Microlithographic Techniques in IC Fabrication, SPIE Vol. 3183, pp. 14-27).  As for the aperture size, always referred to as NA in the manuscript, it is unclear whether the lens diameter or the wavelength or the incidence angle is changed (see eq. 1) as no information is provided.  Finally, the Conclusions section is simply a general discussion which is mostly unrelated to the present study; no summary of important findings or conclusions based on the present study are presented.  Obviously, my difficulties in understanding this study are due to my lack of knowledge within this specific field.  If the intent is to present this work to a limited community, then is is probably acceptable.  However, for a broader audience to understand this work, the abbreviations/jargon need to be replaced by explanatory words.  My evaluation, based on a broader reader base, is that the manuscript is unacceptable in its present form and that a major rewrite is required.   A list of specific comments follows.

At minimum, all abbreviations in the manuscript need a definition when they are first used.   Also, since they all appear to be a short form of either 2 or 3 words, they are probably unnecessary to use at all as they simply inhibit understanding for a large number of readers.  In particular, the beginning of section 2.1 makes extensive use of abbreviations which, for me, made the reading/understanding quite difficult to follow.  Likewise, the captions for figures 2 and 3 would be MUCH easier to understand if, instead of abbreviations, full words were used. 

For similar reasons given above, the caption for Figure 1 does not, for me, provide the necessary information to understand the Figure.

Figures 2 and 3 and 5:  Do the photographs provide any useful information?  If so, what?   Also, text is shown in the photos but is impossible to read.  

I do not follow what is being said in the final paragraph of section 2.1.  Please rewrite.

Section 2.2:  If I understand the first paragraph, the discussion concerns the open area ratio and size.  However, the text uses terms like Cr region and quartz area which is confusing.  Also, it is stated that "when the open area of the mask is 100 nm or less".  I believe one of the dimensions, not the area, is what is meant.

Line 109:  What is meant by "line width roughness"?

Lines 110 and 112 and 114:  "open area ratio" not "open ratio"

Line 114:  Doesn't increasing open area ratio mean the Cr region decreases?

Figure 4:  In part a) a Mass Production Limit is shown without any mention in the text or caption.

Author Response

We appreciate your detailed and helpful review.

This article focuses on making the right PR choice for CD in the ArF Immersion Photolithography process. In order to decrease the line width (CD), the NA must be increased, and at this time, the Depth of Focus (DOF) is decreased. When the PR thickness is sufficiently larger than the DOF, the pattern depth cannot reach the bottom of the PR. This condition causes pattern collapse and poor uniformity in the etching process after the photo process. If the PR thickness is the smallest, there will be no problem in patterning, but the depth margin of the target substrate pattern will be reduced by reducing the etch rate afterwards. Due to this trade-off between NA and DOF, it is essential to select an appropriate PR thickness according to the CD. When the minimum CD is determined in the pattern process, the most suitable (small) NA is set to increase the DOF to the maximum, and the process know-how to maximize the PR thickness is required.

In ArF immersion, the minimum line width is displayed at NA of 0.85 ~ 1.35, which is the NA region of the exposure machine. Measure the DOF along this linewidth. Through this experiment, it becomes a guide to select the PR thickness suitable for the CD. it is confirmed that the resolution changes depending on the Mask Open Ratio in the same NA (Illumination condition), so that the minimum usable minimum line width varies. The last experiment shows that as the DOF and PR thickness approach, the DOF increases due to the reflected light from the PR under-layer.

Jungchul Song 

Reviewer 2 Report

In principle, the relationship between resolution and DOF is a trade-off. This paper experimentally shows the behavior of resolution and DOF according to NA and the mask open ratio in ArF immersion lithography. The experiment was performed faithfully, and the manuscript is robust. However, the following issues must be addressed before publication. 

- Figure 1b & Line #55 in Page 2: Please explain chemically or physically the mechanism by which the optimal energy dose increases with decreasing pattern width.

- Figure 2 & Figure 3: In experiments to determine the effect of NA on DOF, was the exposure dose fixed?

- Figure 5b: Lateral collapse of high-aspect-ratio nanopatterns are found in some SEM images. Lateral collapse interferes with intuitively extracting the correct resolution and DOF. Lateral collapse can be effectively prevented through supercritical CO2 drying (ref: Advanced Functional Materials, 29, 44, 1904971, 2019).

- Why did the authors not use a bottom anti-reflection coating (BARC) in their experiments? Can't you see the standing wave effect in the experimental results?

Author Response

(The authors gave the same response as above.)

Round 2

Reviewer 1 Report

This manuscript, which has been significantly modified from the original version, addresses some of my previous comments.  However, others such as what I find as an extensive use of abbreviations which inhibit understanding by any but those readers directly working in the field and inclusion of photographs containing text that cannot be read have not been addressed, neither within the modified manuscript or in the authors response form that I received.  Thus, I cannot support publication in the present form.  However, should the other referee and the editor feel that the manuscript meets the journal expectations, I will not oppose publication.   Specific comments follow.

Specific comments:

As stated previously, I find the excessive use of abbreviations to severely inhibit the understanding of this manuscript.  Replacement with words would greatly aide in understanding by a much larger audience, plus add only a few additional lines to the entire manuscript.

move definition of CD from line 40 to line 36

Figure 1b:  please add information about the energy density in the caption.  Also, the authors might consider providing a formula for the energy density as a function of numerical aperture as this directly relates to statements in lines 78-79 and lines 121-123.

Figures 2 and 3 and 5:  As stated in my previous review, "Do the photographs provide any useful information?  If so, what?   Also, text is shown in the photos but is impossible to read."

 line 155-56:  Is this sentence written correctly?

Author Response

Dear reviewer 1,

Thank you for your delicate review. I am revising the part you requested. Only one attached file is uploaded, so I'm attaching the review answer. The modified manuscript was delivered to the editor Lebron Tu. 
Thank you again for your review.

Best Regards, 
Jungchul Song

Round 3

Reviewer 1 Report

I thank the authors for their extra efforts in complying with my comments.  Although I assume the previous versions were totally understandable to those familiar with the field my opinion was that to make the article accessible to a wider audience, words rather than abbreviations, were necessary.  I again thank the authors for their efforts and hope that they agree that the present manuscript is improved.  My recommendation is to accept the present version for publication without further review.